# Globally occurring pelagiphage infections create ribosome-deprived cells

Jan D. Brüwer ®[1] ✉, Chandni Sidhu[1], Yanlin Zhao ®[2], Andreas Eich ®[3], Leonard Rößler ®[1], Luis H. Orellana ®[1] & Bernhard M. Fuchs ®[1] ✉

Phages play an essential role in controlling bacterial populations. Those infecting *Pelagibacterales* (SAR11), the dominant bacteria in surface oceans, have been studied in silico and by cultivation attempts. However, little is known about the quantity of phage-infected cells in the environment. Using fluorescence in situ hybridization techniques, we here show pelagiphage-infected SAR11 cells across multiple global ecosystems and present evidence for tight community control of pelagiphages on the SAR11 hosts in a case study. Up to 19% of SAR11 cells were phage-infected during a phytoplankton bloom, coinciding with a ~90% reduction in SAR11 cell abundance within 5 days. Frequently, a fraction of the infected SAR11 cells were devoid of detectable ribosomes, which appear to be a yet undescribed possible stage during pelagiphage infection. We dubbed such cells zombies and propose, among other possible explanations, a mechanism in which ribosomal RNA is used as a resource for the synthesis of new phage genomes. On a global scale, we detected phage-infected SAR11 and zombie cells in the Atlantic, Pacific, and Southern Oceans. Our findings illuminate the important impact of pelagiphages on SAR11 populations and unveil the presence of ribosome-deprived zombie cells as part of the infection cycle.

*Pelagibacterales*, known as the SAR11 clade, are small free-living marine bacteria that account for 20–50% of planktonic cells in the oceans and are crucial components of marine biogeochemical cycles[1]. The reasons for their ecological success in the pelagic ocean are still being elucidated[1]. One proposed explanation was that SAR11 are slow-growing defense specialists, minimally affected by phage predation[2]. However, several phages infecting SAR11 (pelagiphages) have been described[3–7] and studied through cultivation and sequencing efforts with increasing attention in recent years. Metagenomic and -viromic studies not only explored the functional abilities of their genomes[8] but also suggest that pelagiphages, including uncultivated representatives, are the most abundant phages in the ocean[3,9–12]. Despite the ubiquity of pelagiphages, they appear to have low lytic activity within the host population[13,14]. However, direct quantifications of pelagiphage infected cells, and thus investigations

of their role in controlling SAR11 abundance, have not been done so far.

In a recent study, we identified a contrary trend of cell division rates and cell abundances of SAR11 during the 2020 phytoplankton spring bloom at Helgoland Roads, German Bight[15]. Phytoplankton spring blooms are characterized by high phytoplankton-derived organic matter availability and a recurring succession of fast-growing specialized bacterial taxa[16]. Due to low abundances during phytoplankton blooms, SAR11 was generally considered to be outcompeted by specialized taxa. However, when growth rates of SAR11 were measured during the 2020 phytoplankton bloom, SAR11 grew at ~1.9 divisions d$^{-1}$, while cell abundances decreased by ~90% over 5 days[15]. As this decrease was specific to SAR11 (compared to other abundant bacterioplankton), we hypothesized viral-induced mortality to cause the discrepancy[15]. Here, we quantified the number of pelagiphage-

[1]Max Planck Institute for Marine Microbiology, 28359 Bremen, Germany. [2]College of Juncao Science and Ecology, Fujian Agriculture and Forestry University, Fuzhou, China. [3]PSL Research University: EPHE-UPVD-CNRS,UAR 3278 CRIOBE, Moorea, French Polynesia. ✉e-mail: jbruewer@mpi-bremen.de; bruewer_j@gmx.de; bfuchs@mpi-bremen.de

infected cells using advanced microscopy techniques. We first established the protocol on pure cultures of the pelagiphages and their hosts and subsequently assessed infection dynamics throughout the phytoplankton spring bloom described above. For a global perspective, we analyzed the distribution of pelagiphage-infected SAR11 cells in cruise samples across the Pacific, Atlantic, and Southern Ocean. Our investigation of the pure cultures and environmental samples has led us to discover ribosome-deprived but phage-infected cells, a new phenomenon during phage infection.

## Results

### Quantifying phage-infected SAR11 cells and discovery of zombie cells

To characterize pelagiphage:SAR11 interactions, we designed fluorescence in situ hybridization (FISH) probes for the three pelagiphages HTVC027P[10], HTVC031P[4], and *Greip* (*Iscarvirus greipi*; EXVC021P; closely related to HTVC010P)[11,17], that were isolated on the SAR11 strain *Candidatus* Pelagibacter ubique HTCC1062. We targeted those pelagiphages, as they are amongst the most abundant phages globally[3,10] and could be detected in metagenomes originating from the same phytoplankton bloom as described above[18]. These phages are lytic, and a temperate infection can be excluded. We tested hybridization conditions and stringency of the newly designed probes on cultures of *Ca.* P. ubique HTCC1062 infected with either HTVC027P or HTVC031P (Fig. 1). Positive controls with *Greip* were not available to us. As negative controls, we included samples of pelagiphage HTVC023P, which is phylogenetically closely related to HTVC027P[10] but is not targeted by the designed probes (Fig. S1). In a first experiment, we found through FISH and high-throughput image cytometry that 70.0 ± 7.0% (mean ± sd; HTVC027P, $n = 3$) and 17.4 ± 10.1% (HTVC031P, $n = 3$) of the cells

were infected in non-synchronized cultures. The negative control of HTVC023P ($n = 3$) contained <0.1% of false positive signals (Fig. S1, Supplementary Data 1). In an independent second experiment, 36.3 ± 2.7% (mean ± sd) and 32.4 ± 5.0% of cells were infected with HTVC027P ($n = 3$) 18 and 26 h after infection, respectively (Fig. 1). Additionally, 14.3 ± 6.6% and 17.2 ± 6.8% of cells were infected with HTVC031P ($n = 3$) 20 and 28 h post-infection, respectively (Fig. 1).

In all positive controls, we consistently noticed phage-infected cells with no detectable ribosomal RNA signal (22.7 ± 3.1% (t18) and 23.1 ± 3.6% (t26) of total cell counts for HTVC027P; 25.8 ± 17.4% (t20) and 30.4 ± 24.6% (t28) for HTVC031P; Fig. 1, Supplementary Data 1). We named these "zombie" cells as they are probably in a transitional state between living and dead cells. Zombies are different from so-called ghost cells, which were defined as non-living cell envelopes lacking nucleoids[19] or any cytoplasmic content including DNA[20]. In contrast, all zombie cells contained DNA. We excluded the possibility that zombie cells are free phages since they were too large to be individual phages or vesicles according to our image analysis criteria (Supplementary Data 2).

### Phage-infection regulates SAR11 abundance during phytoplankton bloom

To investigate the impact of pelagiphages on the SAR11 host population in the environment, we analyzed 67 samples collected over 133 days in spring 2020 at Helgoland Roads, German Bight. Previously, we showed that fast cell division rates in SAR11 coincided with a rapid decrease in cell abundances (at the end of March and in May; Fig. 2)[15].

We could identify phages as a plausible cause for this unintuitive decrease in cell abundances by quantifying the amount of pelagiphage-infected SAR11 cells in our samples. We found two peaks

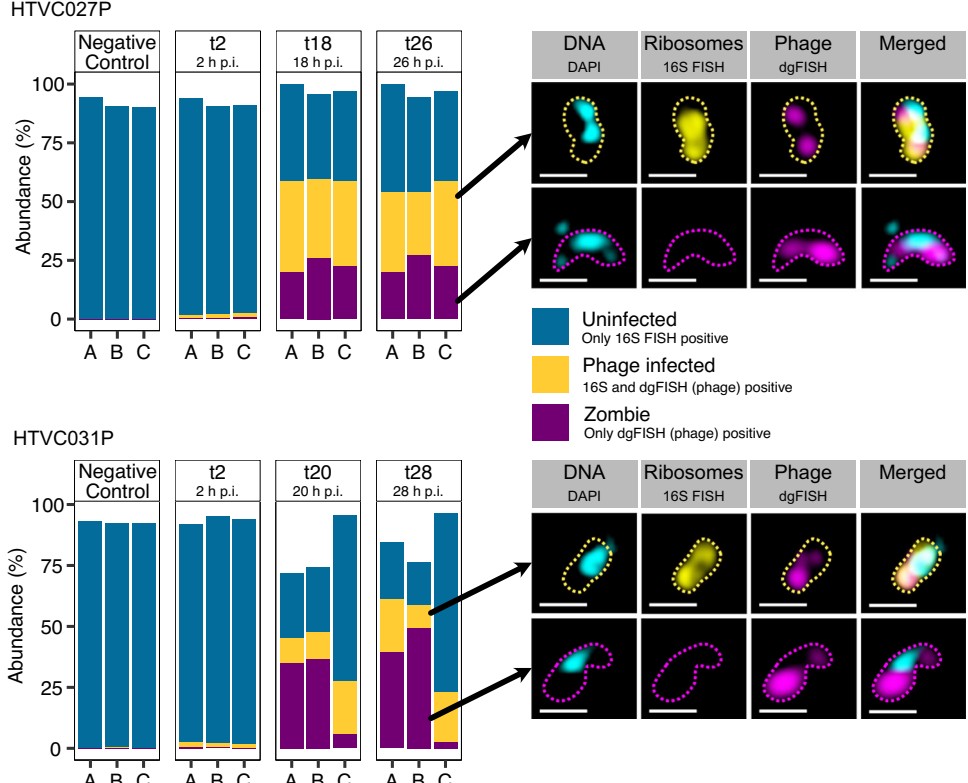

**Fig. 1 | Infection cycle of HTCC1062 infected with HTVC027P and HTVC031P and example epifluorescence microscopy images.** Bargraphs show triplicate samples during the infection cycle. "p.i." stands for post infection. The negative control was uninfected. Abundance of 100% corresponds to total cell counts of DAPI-stained cells. Example microscopy images on the right display DAPI (DNA; cyan), FISH for 16S rRNA (yellow), and phage genes via direct-geneFISH (magenta). Outlines were drawn manually. Images were recorded using SR-SIM on a ZEISS LSM780 equipped with ELYRA PS.1 and analyzed using the ZEN software. Scale bar: 0.5 μm.

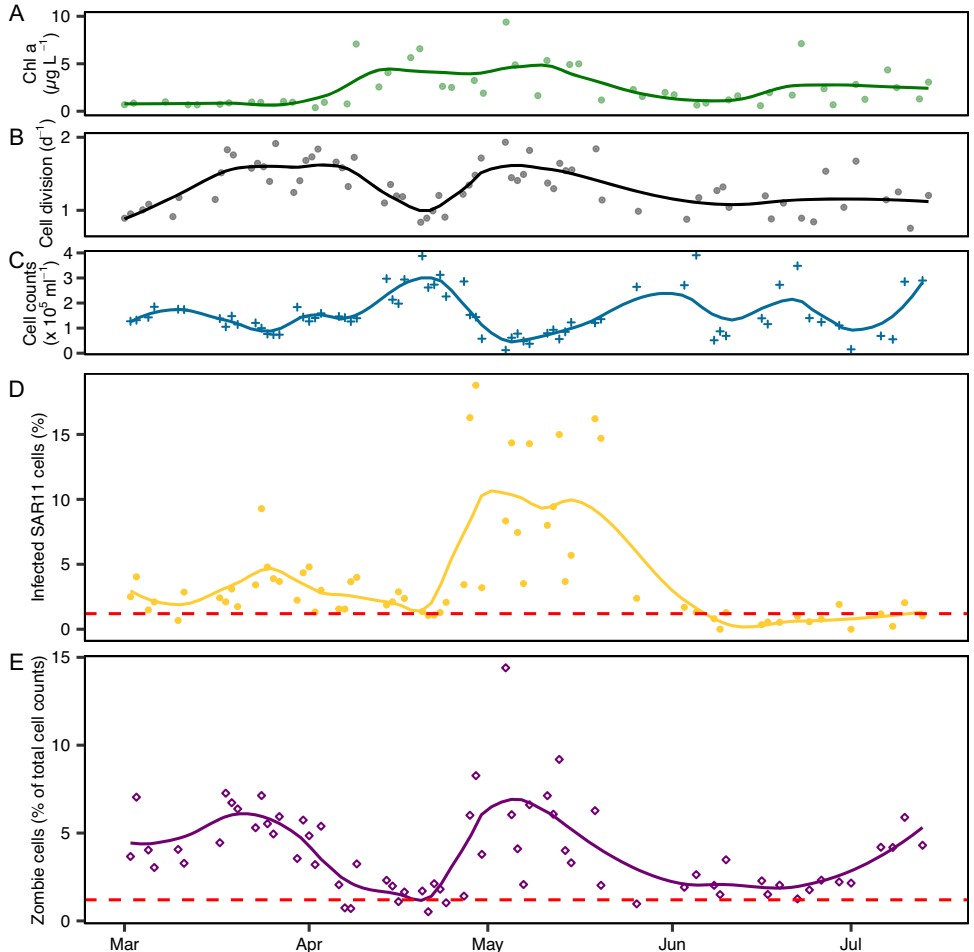

**Fig. 2 | SAR11 and phage population dynamics during 2020 phytoplankton spring blooms at Helgoland Roads. A** Chlorophyll a concentration, as a proxy for phytoplankton bloom development, **B** SAR11 cell division rate, and **C** SAR11 cell count data has previously been presented in ref. 15. **D** Proportion of phage-infected SAR11 cells (yellow; relative to SAR11 cell counts) as raw values per day. Loess smoothing is displayed as line plots. **E** Zombie cells (purple; relative to total DAPI-stained cell counts) are plotted as raw values per day. Loess smoothing is displayed as line plots. The average negative control over all samples is shown as a red-dashed line.

of phage-infected (i.e., ribosome-containing and pelagiphage positive) SAR11 cells, in late March and in early May, accounting for up to 9% and 19% of SAR11 cells, respectively (Fig. 2C). During the remaining sampling period, the abundances of phage-infected cells were close to the detection limit (Fig. 2C) and cell division rates were low (Fig. 2B). These findings highlight the importance of the timing of sampling, potentially explaining why SAR11 phage infection was considered low in earlier studies[13,14,21]. We next assessed the SAR11 community composition in metagenomes from the same phytoplankton bloom sampling campaign in 2020[18]. We determined relative abundances of 16S rRNA gene sequences and metagenome-assembled genomes (MAGs), classified as *Pelagibacterales*, by read mapping. The outcomes from both abundance estimates indicate that the SAR11 community is dominated by the same populations throughout the bloom situation (Fig. 3).

Individual assessments of the three pelagiphages HTVC027P, HTVC031P, and *Greip*, revealed that HTVC031P infected more SAR11 cells than the other two phages during high infection periods. However, differences between the three phages were minor (Fig. S2A), highlighting that infections with each of the three pelagiphages are important in situ. As bioinformatic analyses revealed that the dominating SAR11 species fluctuate simultaneously (Fig. 3) and there are little differences in abundance between the three phages, we believe all species are susceptible to the three assessed phages. Additionally, our findings are in contrast to bioinformatic analyses, that predict

HTVC027P to be more abundant than the other two phages both globally[10,11] and during our sampling period (Fig. S2B; this study). This stresses the importance of experimental evidence for quantification approaches in phage ecology. Differences in abundance estimates between metagenomic and microscopy-based approaches are well-known for bacteria and similar causes may apply for phage abundances. Primer and assembly biases might skew bioinformatic approaches, while low signal intensities might underestimate microscopy-based abundances.

Zombie cell (i.e., ribosome negative but pelagiphage positive) abundances coincided with the number of phage-infected SAR11 cells during the phytoplankton bloom. They were increased at the end of March (max. 7.1% of total cell counts) and early May (max. 14.4%), when phage-infected (i.e., ribosome and pelagiphage positive) SAR11 abundances were highest (Fig. 2). To exclude that the here-assessed pelagiphages cross-infected other bacteria besides SAR11, we visualized all bacteria with the EUB I-III probe[22]. We could not observe a significant difference of zombie abundances (i.e., ribosome negative, but pelagiphage positive) between all bacteria (mean ± sd: $3.4 \times 10^4 \pm 1.6 \times 10^4$ cells ml$^{-1}$) and SAR11 ($4.2 \times 10^4 \pm 2.9 \times 10^4$) over five time-points from 27th April to 4th May (Fig. S3; $F_{(1,1)} = 0.61$, $p = 0.578$, repeated measures ANOVA; Supplementary Data 3). Hence, we conclude that the assessed phages are SAR11-specific. Consequently, the here-observed zombie cells always derived from SAR11 cells.

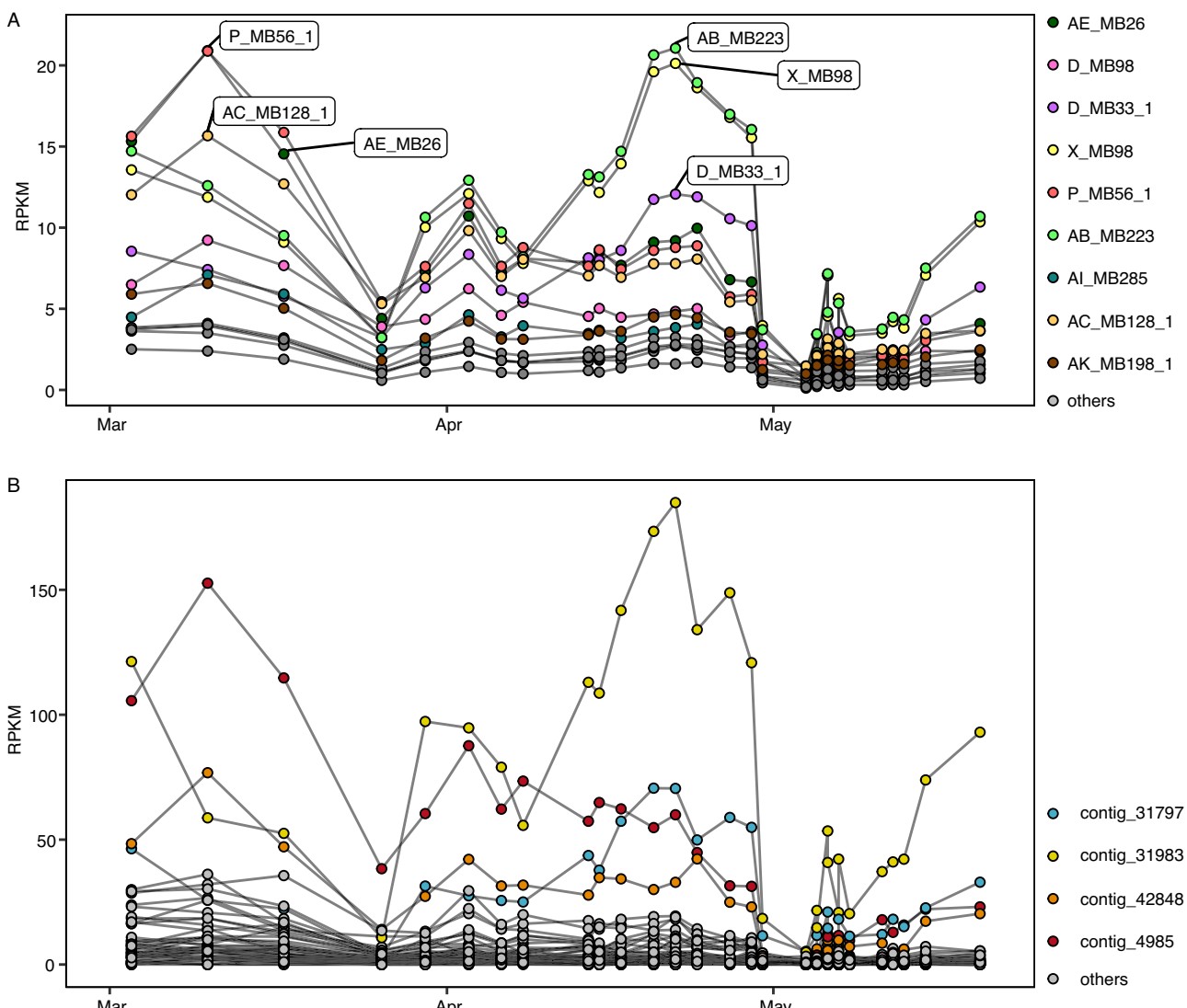

**Fig. 3 | Bioinformatic abundance estimates of SAR11 during phytoplankton spring bloom. A** metagenome assembled genomes (MAG) and **B** 16S rRNA gene sequences classified as Pelagibacterales. All data originated from PacBio Sequel II metagenomes from the 0.2 to 3 μm fraction[18]. RPKM is Reads per Kilobase per Million Mapped reads. $N = 1$ biological sample per data point.

## Global distribution of phage-infected and zombie cells

To assess the broader relevance of our findings, we next examined the global distribution of phage-infected SAR11 and zombie cells in samples collected from surface and deep-chlorophyll maximum water in the Atlantic (cruise PS132[23]; 21 samples from 11 stations; Aug–Sept 2022), Southern (cruise PS133[24]; 22 samples from 11 stations; Oct–Nov 2022), and Pacific Ocean (cruise SO245[25]; 38 samples from 15 stations; Dec 2015–Jan 2016; Fig. 4 and Fig. S4). We detected phage-infected (i.e., ribosome and pelagiphage positive) SAR11 and zombie cells (i.e., ribosome negative but pelagiphage positive) across all three transects (Fig. 4), indicating their global abundance and importance. Phage infection exhibited the lowest prevalence in the Pacific (mean ± sd: 1.3 ± 1.2% of SAR11 cell counts), while the Atlantic (2.9 ± 1.6%) and Southern Ocean (3.3 ± 1.4%) showed higher infection rates. In contrast to phage-infected SAR11 cells, zombie cell abundances were highest in the Pacific (mean ± sd: 5.1 ± 5.1% of total cell counts), followed by lower abundances in the Southern (4.3 ± 1.8%) and Atlantic (2.5 ± 1.2%) Oceans.

SAR11 phage infection and zombie cell abundances were less prominent in the assessed transects compared to the relatively high infection stages observed in the Helgoland Roads phytoplankton bloom data. This suggests that our cruise samples might not have coincided with periods of intense infection by the assessed phages. Nevertheless, across all samples (transects and time-series data), we found a positive correlation between the relative abundance of phage-infected and zombie cells (Fig. S5A; 0% of posterior distribution ≤ 0), a negative correlation between the relative abundance of zombie and SAR11 cells (Fig. S5B; 0% of posterior distribution ≥ 0), and a positive correlation between the relative abundance of phage-infected SAR11 cells and the frequency of dividing cells, which is a proxy for cell division activity (Fig. S5C; 0.000125% of posterior distribution ≤ 0)[15]. This indicates higher infection rates in faster-growing hosts. Our microscopic evidence of phage-infected and zombie cells indicates that they are globally distributed and suggests that zombie cells are an integral part of pelagiphage infections.

## Discussion

In this study, we used direct-geneFISH to visualize and quantify the abundance of phage-infected SAR11 cells. We assessed the impact of phage infection on the SAR11 community during a phytoplankton bloom and showed the global distribution of phage infections by the assessed phages. We additionally discovered ribosome-deprived but

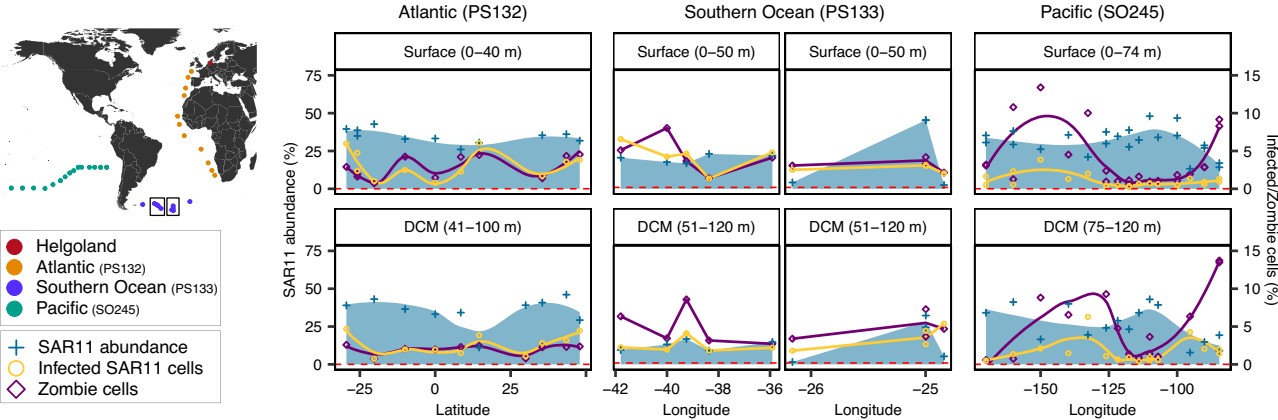

**Fig. 4 | Global distribution of SAR11, phage-infected SAR11, and zombie cells.** Map with sampling locations during different cruises and long-term ecological research station Helgoland Roads. Relative abundance of SAR11 (left y-axis), phage-infected SAR11 (right y-axis), and zombie cells (right y-axis) from the Atlantic, Southern Ocean, and Pacific with respective cruises in brackets. Raw data is displayed as individual points and areas represent loess smoothing for the Atlantic and Pacific. Subset (boxes) of data from Southern Ocean without smoothing is displayed. Complete data of the Southern Ocean is available in Fig. S4. $N = 1$ sampled station per shown datapoint.

phage-infected cells, whose relative abundances are correlated to the amount of phage-infected SAR11 cells.

During the phytoplankton bloom, up to 19% of SAR11 cells were phage-infected, which cooccurred with a ~90% decrease in SAR11 abundances. Assuming in situ phage lysis within 24 h or faster[5], our results are in line with global estimates of up to 30% phage-mediated cell lysis within 1 day[12,26]. We interpret this as part of a boom and bust cycle, where SAR11 cell abundances reached a critical population density that was subsequently reduced by phage infection. The "kill-the-winner" hypothesis describes the disproportionally higher phage-induced lysis rates of faster-growing bacteria. Further, different bacterial strains from the same species may succeed each other, especially in bloom situations, as the majority of phages are believed to be strain-specific[27,28]. As metagenomic data suggest that the SAR11 community was dominated by the same population throughout the phytoplankton bloom[18] (Fig. 3), our findings of fast-growing SAR11 and high infection rates suggest a kill-the-winner behavior[28]. In addition, the almost equal contribution of phage infection by the three assessed phages suggests that all the abundant strains are susceptible to the assessed phages. The dominance of individual strains succeeding each other would be in accordance with previously reported Red Queen dynamics, describing constantly changing host and viral communities on the fine strain level[29].

## Zombie cells: clever persisters or phage puppets?

Pelagiphage-infection of SAR11 cells results in the formation of zombie cells – cells devoid of any detectable 16S ribosomal RNA. Such a major cell transformation may be caused by a yet-unknown anti-phage defense system. Infected cells may digest their ribosomes to prevent phage proteins from being synthesized and reduce their metabolism—they may become persisters as recently proposed[30]. In fact, we have previously shown the ability of SAR11 to modify their ribosome content according to growth rates[15]. However, no anti-phage defense system targeting rRNA or an abortive infection has been described for SAR11 or any other bacterium yet[31]. Nevertheless, we screened all SAR11 MAGs ($n = 14$), which could be recovered from the phytoplankton bloom in 2020, none of which contained any anti-phage system (Supplementary Data 4). We further analyzed 172 publicly available *Ca.* Pelagibacter ubique genomes, of which 19 contained putative anti-phage systems. These were all classified as restriction-modification (RM) systems and are involved in epigenetic modifications (i.e., methylation or phosphorothiolation of the ribophosphate) of the host DNA and restrictions of un-modified phage-DNA[32] (Supplementary

Data 4). However, a close monitoring of HTVC027P and HTVC031P infections of batch-cultured HTCC1062 over time speak against an anti-phage system. Firstly, if an anti-phage system existed, zombie cells would enrich over time, while the phage would eventually lyse 16S rRNA-containing infected cells. We could not detect abundance differences 18 and 26 h (HTVC027P) or 20 and 28 h (HTVC031P) post-infection for both cell types ($t(3.9) = -0.166$, $p = 0.88$ for HTVC027P, $t(3.6) = -0.26$, $p = 0.81$ for HTVC031P, Welch's two sample t-test, Fig. 1). Additionally, we assessed the phage DNA content by measuring the phage FISH fluorescence intensity. The fluorescence intensity in cells infected with HTVC027P increased over time ($F(1) = 0.31$, $p < 0.001$) indicative for ongoing phage DNA synthesis, but we could not detect differences between zombie and phage-infected 16S positive cells ($F(1) = 0.303$, $p = 0.582$, ANOVA, Fig. S6). For cells infected with HTVC031P, the fluorescence not only increased over time, but was also increased in the zombie cells, compared to the phage-infected 16S positive cells ($F(1) = 100.18$, $p < 0.001$). This indicates a continuous production of phage DNA in both, the zombie and the infected 16S rRNA containing cells and speaks against a defense mechanism. Lastly, the host's DNA content, and consequently their abundance in metagenomes across the 2020 spring bloom, would remain unchanged, if zombies were a result of an anti-phage system. However, SAR11 cell abundances from FISH and metagenomic (MAG and 16S rRNA-based) data do fluctuate synchronously (Figs. 2, 3), rendering the hypothesis of an anti-phage defense in SAR11 unlikely.

Alternatively, phage infection may result in phage-induced RNA degradation to recycle ribonucleotides for phage genome synthesis. In this case, the phages become the puppet masters of the SAR11 host[26]. The use of host RNA to build new phage genomes has previously been suggested for *Prochlorococcus* cyanophages[33] and is not limited to rRNA but may also include mRNA, tRNA, or other RNAs. In fact, we detected all intermediate stages between phage-infected with and without a ribosomal signal, suggesting the general use of RNAs (Fig. S7). In the pure cultures of HTVC027P, we assessed the 16S FISH signal areas, which were truncated at the lower detection limit, indicating a continuum of decreasing ribosomal content (Fig. S7). Hence, there is no switch (i.e., "yes" or "no") whether ribosomal RNA is used during phage infection but rather suggests a continuous use of all available RNA to complement the nucleotide pool.

The annotation of HTVC031P revealed the presence of ribonucleases and deoxyribonucleotide dehydrogenase subunits alpha and beta (Supplementary Data 5), which are essential for the breakdown of

RNA and the subsequent conversion to DNA nucleotides. On the genome, the ribonucleases and dehydrogenases were middle genes, located and transcribed between DNA replication (early) and structural (late) phage proteins encoding genes[34]. Timing is essential, as nucleotides must be available early in the infection cycle, while phages rely on host ribosomes, which should not be digested too early. In the case of HTVC027P and *Greip*, the dehydrogenase genes were not found in their genomes. Either host proteins or one of the many uncharacterized proteins (Supplementary Data 5) may be used. RNA nucleotides are a valuable resource, especially when DNA nucleotides are scarce. Nucleotides are frequently a limiting factor. For example, cyanophages not only enhance their host's photosynthesis activity but also modify the host's metabolism to increase nucleotide production[35], although the host genomes are much larger than SAR11. SAR11 have amongst the smallest genomes (~1.3 million base pairs) of free-living bacteria[1], and they might have epigenetic modifications in their DNA (discussed above), reducing the availability of DNA nucleotides[36]. At the same time, SAR11 have an estimated 150–700 ribosomes per cell[37], which equates to 0.6–3.0 million bases of single-stranded RNA. This could be a valuable resource, almost doubling the number of nucleotides available for double-stranded DNA phage genomes. To summarize, while we cannot entirely exclude an anti-phage system as a cause for zombie cells, we believe phage-induced ribosome digestion is more likely.

We present the first microscopic quantification of phage-infected SAR11 cells in situ, advancing our understanding of SAR11 clade dynamics, and present zombie cells as a new phenomenon. We demonstrate that pelagiphage infections play a critical role in regulating SAR11 populations, especially during phytoplankton spring blooms. The global prevalence, evident by our data from various oceans, highlights their significance in marine microbial ecology. The discovery of the globally occurring zombie cell phenomenon underscored the complexity of phage-host interactions. We provide possible explanations for the formation of zombie cells, which are phage-infected cells without detectable rRNA. We suggest a phage-induced recycling of ribosomal RNA, though this requires further exploration in future studies. The observed zombie cells are a result of phage infection of SAR11. However, as other phage-host pairs likely result in similar phenomena, our discovery of zombies has implications beyond marine microbial ecology. This research not only sheds new light on the intricate dynamics of SAR11 and their viruses, as well as their turnover rates, but also opens new possibilities for exploring microbial and viral strategies in the ocean's biogeochemical cycles.

## Methods

### Cultivation of *Candidatus* Pelagibacter ubique infected with HTVC031P, HTVC027P, and HTVC023P

The SAR11 strain *Candidatus* Pelagibacter ubique HTCC1062 was kindly provided by Professor Stephen Giovannoni, Oregon State University, USA. HTCC1062 was cultured in artificial seawater-based ASM1 medium supplemented with 1 mM $NH_4Cl$, 100 μM $KH_2PO_4$, 1 μM $FeCl_3$, 100 μM pyruvate, 50 μM glycine, and 50 μM methionine[38]. HTCC1062 cultures were incubated at 17 °C without shaking and light. Exponentially growing HTCC1062 cultures were infected with HTVC023P, HTVC027P, and HTVC031P independently at a phage-bacteria ratio of ~10:1 in triplicates. Cell mortality was monitored using the Guava EasyCyte flow cytometer (Millipore, USA). When cell mortality was detected, samples were fixed with formaldehyde (1% final concentration) for 1 h at room temperature and filtered on 0.2 μm polycarbonate filters (Merck Millipore, Burlington Massachusetts, US). In a repeated experiment, *Ca*. P. ubique HTCC1062 were infected with HTVC027P and HTVC031P, as described above. Samples were taken from three time-points (2 h after infection; ~2 h before cell lysis; ~6 h after cell lysis) and fixed and filtered as described above.

### Environmental sampling

Samples were collected during the 2020 phytoplankton spring bloom from -1 m depth at the long-term ecological research station Helgoland Roads (54° 11.3′ N, 7° 54.0′ E), German Bight[15] (Supplementary Data 6). Samples from the Atlantic and Southern Ocean were collected using a Seabird SBE 911 + CTD in 2022 during the R/V Polarstern cruises PS132[23] and PS133/1[24], respectively[39]. Samples from the Pacific Ocean were collected with a Seabird SBE 911 + CTD during the R/V Sonne cruise SO245[25]. SAR11 cell counts from SO245 were retrieved from ref. 40 During the cruises, samples were collected from surface water and deep-chlorophyll maximum (DCM; Supplementary Data 6). Samples were fixed with formaldehyde (1% final concentration) for 1 h at room temperature. Cells were immobilized on 0.2 μm polycarbonate filters (Merck Millipore, Burlington Massachusetts, US), which were stored at −20 °C until further processing. The final sampling volume varied depending on total cell counts in the samples (Supplementary Data 6).

### Pelagiphage FISH probe design and synthesis

We designed direct-geneFISH[41] probes based on alignments between each of the three isolates, namely HTVC027P, HTVC031P, and *Greip*, and PacBio Sequel II metagenomes from the 2020 spring phytoplankton bloom at Helgoland, North Sea[18] (ENA PRJEB52999). Target probe-regions were identified from assembled contigs (Supplementary material and methods). Subsequently, probes were designed manually within Geneious (v2022.1.1)[42]. We aimed for 10–13 probes per phage of 156–318 bp length and a GC content similar to the phages (22.0–43.2%, mean ± sd: 32.9 ± 4.9%). Further, reference genomes and metagenome data from Helgoland Roads needed to share a minimum of 90% nucleotide identity for usability during FISH[43]. We aimed to target genes encoding terminases, polymerases, or structural proteins, as we expect higher conservancy in these genes. Ambiguous alignment with any other sequence was excluded, against the nt database using the NCBI BLAST webservice (February 14th 2023).

Probes (Supplementary Data 7) were ordered as "oPools" from integrated DNA Technologies (IDT, Coraville, Iowa, USA) and resuspended in water as directed by the manufacturer. Probes were labeled with the ULYSIS Alexa 594 conjugation kit (Invitrogen, Waltham, Massachusetts, USA) with double the suggested fluorophore concentration, as this successfully enhances the labeling efficiency[43]. Labeled probes were subsequently purified using Micro Bio-Spin chromatography columns P-30 (Bio-Rad, Hercules, California, USA). Labeling efficiencies were calculated as described by Invitrogen, using a NanoDrop (Thermo Fisher Scientific, Waltham, Massachusetts, USA).

### Fluorescence in situ hybridization

First, CARD-FISH targeting the 16S rRNA of SAR11 ("SAR11-mix", Supplementary Data 7) was conducted (details see Supplementary Methods)[44]. Secondly, samples were hybridized with equimolar amounts of the probes targeting HTVC027P, HTVC031P, and *Greip*, using direct-geneFISH as described earlier (details see Supplementary Methods)[41,43] with minor modifications. Hybridization buffer with 25% formamide was used and no ethanol washing was conducted after FISH to prevent any loss of fluorescence signal. Hybridized filters were counter-stained with the DNA stain 4′,6-diamidino-2-phenylindole (DAPI; 1 μg ml⁻¹). Samples were embedded in ProLong Glass Antifade (Invitrogen, Waltham, Massachusetts, USA) for microscopy. As negative controls, samples of HTVC023P were hybridized with the probe mix, targeting all three phages. Additionally, environmental negative controls included samples which were not exposed to the phage-probe mix to account for any autofluorescence within cells.

### Microscopy

Samples were imaged on a Zeiss AxioImager.Z2m, equipped with a charged-coupled device (CCD) camera (Zeiss AxioCam MRm, Zeiss,

Oberkochen, Germany), and illuminated with a Zeiss Colibri 7 LED (excitation: 385 nm for DNA, 469 nm for 16S rRNA CARD-FISH, and 590 nm for direct-geneFISH signals). The microscope was equipped with a multi-Zeiss 62 HE filter cube (Beam splitter FT 395 + 495 + 610). Images were recorded with a custom-built macro[45,46] within the Zeiss AxioVision software (Zeiss, Germany). A total of 120 fields of view per sample were recorded with a 63x Plan Apochromat objective (1.4 NA, oil immersion). For high-resolution imaging, we used a Zeiss LSM 780 (Zeiss, Oberkochen, Germany), with an ELYRA PS.1 detector upgrade. The microscope was equipped with a 63x plan apochromatic oil immersion objective and the excitation lasers 405 nm (DAPI), 488 nm (16S rRNA CARD-FISH), and 591 nm (direct-geneFISH).

### Image cytometry

Quality control and automated cell counting of 8-bit greyscale images was done within the Automated Cell Measuring and Enumeration tool (ACME, available from https://www.mpi-bremen.de/automated-microscopy.html)[45,46] with channel-specific settings (Supplementary Data 2). Cells for total cell counts were defined by a DNA (DAPI)-specific signal. SAR11 cells were defined with an overlapping DNA and 16S rRNA (CARD-FISH) signal and phage-infected cells needed an additional phage (direct-geneFISH) signal. Zombies were cells with a phage signal but no 16S rRNA signal. We calculated the frequency of dividing cells—a proxy for cell division rate—as previously described[15] using the MicrobeJ plugin[47] within Fiji/ImageJ[48]. In principle, a cell containing two local DNA maxima was counted as a dividing cell.

### Metagenomic abundance estimates for SAR11 MAGs, 16S rRNA gene, and Pelagiphages

To determine the relative abundance of SAR11 metagenome-assembled genomes (MAGs) during the 2020 phytoplankton spring bloom, we performed a mapping analysis utilizing PacBio metagenomic reads obtained from the prokaryotic fraction (0.2–3 μm) across all 30 samples. The reference MAGs, classified under the order *Pelagibacterales* by gtdb-tk (v1.3.0, release 202)[49], were initially derived from the same phytoplankton bloom metagenomes, described above[18]. Raw reads were mapped using the minimap2-pb[50] algorithm, executed within the SqueezeMeta pipeline (version 1.3.1)[51]. The mapping outcomes were normalized using the reads per kilobase per million mapped reads (RPKM) metric, which considers both the length of the MAG and the library size of each sample. The RPKM value was determined using the formula $RPKM = \frac{\text{Reads mapped to SAR11 MAG}*10^6}{\text{total read in a sample*length of MAG in kilobase pairs}}$.

For quantifying the abundance of the 16S rRNA gene, we extracted the full-length 16S rRNA sequences from metagenome assemblies using Barrnap (version 0.9; https://github.com/tseemann/barrnap). Similar to the SAR11 MAGs, these sequences underwent mapping, and their relative abundance was computed using the RPKM method as described earlier.

To assess the relative abundance of phages HTVC027P, HTVC031P, and *Greip*, mapping of metagenome reads to the reference genomes of these phages were performed in the similar fashion. To facilitate a comprehensive comparison with our microscopy data, and considering the specificity of these phages to SAR11, we calculated the abundance of these *Pelagiphages* relative to SAR11 community present during spring phytoplankton bloom in 2020. Therefore, phage relative abundance was determined as $\frac{\text{Reads mapped to phage genome}*10^6}{\sum \text{read mapped to all SAR11 MAGs*length of phage in kilobase pairs}}$.

### Identification of defense systems within SAR11 genomes

All available *Pelagibacter ubique* genomes (n = 172) available in the RefSeq database (from October 22nd, 2023)[52] and MAGs from the 2020 phytoplankton spring bloom (n = 14)[18] were screened for anti-phage defense mechanisms. DefenceFinder[53] was used with the default database from October 22nd, 2023.

### Statistics and modeling

Statistical analyses and corresponding visualizations were done in R (v4.2.2)[54] (for used packages see supplementary material and methods). Repeated measures ANOVA was used to test for the specificity of pelagiphages to SAR11. Zombies were detected in all bacteria, using the 16S FISH probe EUB I-III, and compared to zombies in SAR11 (SAR11-mix, Supplementary Data 7). Samples originated from the time series and were not independent from each other. Thus, repeated measures ANOVA was chosen.

Bayesian beta regressions were applied to assess the relationship between (a) the abundance of phage-infected and zombie cells, with the model formula rel_infT ~Zombie_cells and (b) abundance of zombie and SAR11 cells, using the model formula Zombie_cells ~ rel_SAR11_abundance. rel_infT is the relative infection rate, transformed by adding 0.001 because two values of the data (148 data points) originally contained 0 for which the beta distribution is not defined; Zombie_cells is the relative abundance of Zombie cells; rel_SAR11_abundance is the relative SAR11 abundance; and FDC_percent is the frequency of dividing cells. The model predictions were back-transformed from the logit-scale for the plots in Fig. S5.

a. We assumed a positive relationship between phage-infected cells and Zombie cells, as the 95% credible interval [4.083, 9.554] of the slope (on logit-scale) excluded 0 and all values of the posterior distribution for the slope were ≥ 0. The model predicts an intercept of −3.72 ± 0.1 and a slope of 6.92 ± 1.41 (on logit-scale; means ± SD).

b. A negative relationship was assumed between the relative abundances of zombie and SAR11 cells, as all values of the posterior distribution for the slope were <0. The 95% credible interval was [−2.800, −0.953] (on logit-scale). The model predicted an intercept of −2.74 ± 0.1 and a slope of −1.87 ± 0.5 (on logit-scale; mean ± SD).

c. We assumed a positive relationship between phage-infected cells and Zombie cells, as the 95% credible interval [0.02, 0.05] of the slope (on logit-scale) excluded 0 and only 1 of the 8000 values of the posterior distribution for the slope was ≤0. The model predicts an intercept of −3.72 ± 0.1 and a slope of 0.04 ± 0.01 (on logit-scale, mean ± SD).

In the model, flat priors (brms default) were used and 2000 iterations for 4 chains after a warmup period of 2000 iterations per chain.

### Reporting summary

Further information on research design is available in the Nature Portfolio Reporting Summary linked to this article.

## Data availability

The microscopy data generated in this study have been deposited in the Edmond database [https://doi.org/10.17617/3.3ZLOAT]. Metagenomes assessed in this study are publicly available from the European Nucleotide Archive (ENA) with the accession code PRJEB52999. In this manuscript, the following databases were used: GTDB release 202 [https://data.gtdb.ecogenomic.org/], RefSeq (online via NCBI website) as of October 22nd, 2023 [ftp://ftp.ncbi.nlm.nih.gov/refseq/release/viral], and the nt database of the NCBI blast server as of February 14th, 2023.

## Code availability

Code to create figures and statistical analyses have been uploaded to GitLab: <https://gitlab.mpi-bremen.de/jbruewer/pelagiphage-abundance>.

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

## Acknowledgements

We thank Prof. Dr. Rudolf Amann for general support and critical feedback on different stages of the manuscript and Dr. Susanne Erdmann for general discussions. Germinate Science is acknowledged for language editing and Anja Greiser for help with FISH analysis. We would like to acknowledge the captains, crews, and scientific personnel on the R/V Polarstern cruises PS132 and PS133/1, as well as R/V Sonne cruise SO245. PS132 was financed by the grant AWI_PS132_01 and PS133 by grant AWI_PS133/1_06 by the Alfred Wegener Institute. Cruise SO245 was financed by grant 03G0245A of the Federal Ministry of Education and Research of Germany. Funding was provided by the German Research Foundation (DFG) project FOR 2406 'Proteogenomics of Marine Polysaccharide Utilization (POMPU)' by grants to BMF (277249973), by the National Science Foundation of China (NSFC) under grant number 4207610 to Y.Z., and by the Max Planck Society to J.D.B., L.R., C.S., L.H.O., B.M.F.

## Author contributions

J.D.B. conceptualized and designed study, generated and analyzed data, interpreted results, created figures, wrote and revised manuscript; C.S. analyzed data; Y.Z. provided samples and interpreted results; A.E. analyzed data; L.R. helped during sampling and generated data; L.H.O. analyzed data; B.M.F. conceptualized study, interpreted results, and revised manuscript. All authors reviewed the manuscript and provided feedback.

## Funding

## Competing interests

The authors declare no competing interests.
