## [Peer Review File · Nature Communications]

Globally occurring pelagiphage infections create ribosome-deprived cellsEditorial Note: This manuscript has been previously reviewed at another journal that is not operating a transparent peer review scheme. This document only contains reviewer comments and rebuttal letters for versions considered at Nature Communications.

Reviewers' Comments:

Reviewer #1:

Remarks to the Author:

I found the authors' comments and revisions satisfactory and have no additional comments.

Reviewer #3:

Remarks to the Author:

This is a much improved version, and the authors have answered the large majority of my questions. I still have a few concerns (more minor than before):

Line 51. The statement that the decrease was taxon-specific needs more explanation. It was not even obvious to me after looking at the cited paper. Is this meant to say SAR11 and only SAR11 declined so much while other similar sized organisms did not, so might imply grazing was an unlikely contributor to the loss?

Fig 1. This better supports the authors' conclusions, but I still wonder what happened to yield the previous images I had commented on that gave a very different impression (see prior review). It makes me worry a bit about "cherry picking" supportive images.

Lines 113-116 and Fig 3 re SAR11 strains staying the same.: It is very hard to use read mapping to be certain that strains do not change, especially when only the average coverage is shown. Read mapping cannot show anything about "new" genes present in environmental genomes but lacking in the reference genomes. And did a close look at the read maps show consistent coverage along the entire genome, or were there islands (indicative of strain differences) ? It would not show up unless you look closely at the coverage maps (e.g. with anvio). And 16S data is too phylogenetically coarse to detect strain differences so you can't really rely on that. Also - MAGs notoriously miss most SAR11 organisms because of microdiversity breaking up the contigs. Big marine MAG collections have a tiny fraction in SAR11 even though the group represents about a third of the organisms, so those making MAGS are highly atypical.

Fig. 2. Panel D is potentially confusing or even misleading. The same scale is used for % infected SAR11 only and for % of all cells being zombies. But SAR11 is only about a third of cells, so putting them on the same graph with the same scale could easily give a false impression. I suggest two graphs with clearly labeled axes (even though the legend explains – easy to miss the nuance).

Line 151. but by Fig 2 it looks like Zombies exceed infected SAR11, especially given the percentage of infected cells is SAR11 only and Zombies is all cells. Seems to be a discrepancy needing explanation. see below

Lines 162-165. In the Pacific only 1% of SAR11 cells alone were infected, but 5% of all cells (e.g. about 15 times the number of infected SAR11 cells) were zombies. Begs for an explanation here – e.g.

many or probably the large majority of zombies are clearly not infected SAR11 cells. Other species making zombies? If so why report the totals when it is comparing "apples and oranges"? Seems potentially misleading without clear explanations.

Line 194 – Here saying zombies are correlated to phage infected SAR11 again seems misleading when in the Pacific there are about 15 X as many zombies as infected SAR11. Really need to be up front about this and what it means.

Line 216-217. Reporting that these authors previously showed SAR11 regulate RNA content seems to be a stretch of the facts. When I looked at the previous paper it showed ribosome content was indeed higher per cell in some samples, but tellingly in those exact same samples the cells were much larger (especially considering biovolume varies as the cube of diameter). So it could just be that ribosomes are a particular fraction of cytoplasm and therefore it may not be regulation of RNA at all but simply more cytoplasm having more ribosomes. And faster growing cells are bigger. I think the claim of specific RNA regulation should be removed.

Lines 286-287. While this could explain why there are so many more zombie cells than infected SAR11 it should be clarified throughout the manuscript when comparing total zombies and fraction of infected SAR11. The rest of the manuscript implies it is all about SAR11, by omission of this discussion.

Lines 407-409. defense mechanisms are easily lost in MAG production because they often have different tetranucleotide frequencies than the underlying core genome and also not all members of a species have them.

Should add a caveat about this.

Comments by the reviewers are indicated in **black**.
Responses by the authors are indicated in **blue**.
New and revised manuscript sections are indicated in **green**.

REVIEWERS' COMMENTS

Reviewer #1 (Remarks to the Author):

I found the authors' comments and revisions satisfactory and have no additional comments.

We thank the reviewer for their valuable time to evaluate our revised manuscript.

Reviewer #3 (Remarks to the Author):

This is a much improved version, and the authors have answered the large majority of my questions.

Thank you very much for your valuable feedback. Your previous and current comments have helped us to considerably improve our manuscript.

I still have a few concerns (more minor than before):

Line 51. The statement that the decrease was taxon-specific needs more explanation. It was not even obvious to me after looking at the cited paper. Is this meant to say SAR11 and only SAR11 declined so much while other similar sized organisms did not, so might imply grazing was an unlikely contributor to the loss?

Yes. This is correct and this is the fascinating observation: SAR11 declined much more in abundance than the other assessed bacterial groups. They also had the smallest cell sizes. It is generally accepted in microbial ecology that small cell sizes reduces the risk of being preyed upon by grazers. Consequently, the most plausible explanation to the SAR11 - specific decline is a viral-induced mortality.

Fig 1. This better supports the authors' conclusions, but I still wonder what happened to yield the previous images I had commented on that gave a very different impression (see prior review). It makes me worry a bit about "cherry picking" supportive images.

Thank you for your comments. We took these very serious. As a consequence, we have analysed 120 908 cells (in 555 field of views; HTVC027P) and 44 818 cells (in 636 field of views; HTVC031P) and quantified the DNA (DAPI) and phage (direct-geneFISH) content. It turned out that the differences between phage-infected, ribosome-containing and zombie cells were negligible (Fig. S6). Due to the spread of fluorescence signals it is challenging to choose representative images, but we did our best to find suitable images and hope these will be convincing now in the changed figures. For full transparency, all raw images have been provided in the online repository.

Lines 113-116 and Fig 3 re SAR11 strains staying the same.: It is very hard to use read mapping to be certain that strains do not change, especially when only the average coverage is shown.

Read mapping cannot show anything about “new” genes present in environmental genomes but lacking in the reference genomes. And did a close look at the read maps show consistent coverage along the entire genome, or were there islands (indicative of strain differences) ? It would not show up unless you look closely at the coverage maps (e.g. with *anvi'o*). And 16S data is too phylogenetically coarse to detect strain differences so you can't really rely on that. Also - MAGs notoriously miss most SAR11 organisms because of microdiversity breaking up the contigs. Big marine MAG collections have a tiny fraction in SAR11 even though the group represents about a third of the organisms, so those making MAGS are highly atypical.

All MAGs analyzed in our study were recovered from long-read PacBio HiFi metagenomes. Long-read metagenomes can help resolve complex genomic regions, including repetitive elements and structural variations, thereby reducing ambiguity associated with strain heterogeneity. Additionally, all MAGs were manually refined using *anvi'o* through visual inspection and consideration of coverage across samples, further reducing the uncertainty linked with strain diversity. Our close inspection of coverage values across genes in the assessed MAGs suggest that although it's possible to have multiple strains within a single MAG, they're unlikely to shift in abundances over time. This suggests that the abundance pattern we observed for a particular MAG arise from a homogeneous community.

Regarding the 16S rRNA gene, while acknowledging its limited resolution in discriminating closely related taxa or resolving fine-scale phylogenetic relationships, we emphasize the value of employing high-quality, full-length sequences. In Figure 3, the inclusion of full-length 16S rRNA gene sequences (1470 bp in length) from PacBio HiFi metagenomes enhances the reliability of depicting genetic diversity.

In the manuscript, we intended to emphasize that the SAR11 populations and community composition did not change a lot. Due to the known challenges with SAR11 MAGs, we replaced the word ‘species’ and ‘strains’ by ‘populations’ throughout the manuscript.

Fig. 2. Panel D is potentially confusing or even misleading. The same scale is used for % infected SAR11 only and for % of all cells being zombies. But SAR11 is only about a third of cells, so putting them on the same graph with the same scale could easily give a false impression. I suggest two graphs with clearly labeled axes (even though the legend explains – easy to miss the nuance).

We have changed the figure accordingly.

Line 151. but by Fig 2 it looks like Zombies exceed infected SAR11, especially given the percentage of infected cells is SAR11 only and Zombies is all cells. Seems to be a discrepancy needing explanation. see below

We acknowledge that this might have been confusing. In this experiment we show that zombie cells were always derived from SAR11 cells and that no non-SAR11 cells were infected by the assessed pelagiphages (as written in the text, relevant data can be found in supplementary figure S3, not in figure 2). Therefore, there is no discrepancy, as phage-infected cells (pelagiphage containing and ribosome positive) are different from zombie cells (pelagiphage positive but ribosome negative). To avoid the confusion, we changed the wording throughout the manuscript to make the differentiation clear.

Here, we changed the text to:

We could not observe a significant difference of zombie abundances (i.e., ribosome negative, but pelagiphage positive) between all bacteria (mean \pm sd: $3.4 \times 10^4 \pm 1.6 \times 10^4$ cells ml⁻¹) and SAR11 ($4.2 \times 10^4 \pm 2.9 \times 10^4$) over five time-points from 27th April to 4th May (Fig. S3; $F(1,1)=0.61$, $p=0.578$, repeated measures ANOVA; table S3). Hence, we conclude that the assessed phages are SAR11-specific. Consequently, the here-observed zombie cells always derived from SAR11 cells.

Lines 162-165. In the Pacific only 1% of SAR11 cells alone were infected, but 5% of all cells (e.g. about 15 times the number of infected SAR11 cells) were zombies. Begs for an explanation here – e.g. many or probably the large majority of zombies are clearly not infected SAR11 cells. Other species making zombies? If so why report the totals when it is comparing “apples and oranges”? Seems potentially misleading without clear explanations.

Thank you for this comment. All zombie cells originally derived from ‘healthy’ (i.e., uninfected SAR11) cells. However, they are not identified as SAR11 cells by our 16S rRNA targeted probes anymore due to the lack of their ribosomes. We have rephrased parts of the manuscript to make the differentiation clearer.

Line 194 – Here saying zombies are correlated to phage infected SAR11 again seems misleading when in the Pacific there are about 15 X as many zombies as infected SAR11. Really need to be up front about this and what it means.

We do acknowledge that the numbers are not identical, but the relative abundances of pelagiphage infected SAR11 and zombie cells do correlate, as shown in Fig. S5. This correlation is apparent over all data (from all sampling campaigns) and also, when focusing on samples from the Pacific only.

Line 216-217. Reporting that these authors previously showed SAR11 regulate RNA content seems to be a stretch of the facts. When I looked at the previous paper it showed ribosome content was indeed higher per cell in some samples, but tellingly in those exact same samples the cells were much larger (especially considering biovolume varies as the cube of diameter). So it could just be that ribosomes are a particular fraction of cytoplasm and therefore it may not be regulation of RNA at all but simply more cytoplasm having more ribosomes. And faster growing cells are bigger. I think the claim of specific RNA regulation should be removed.

The reviewer is correct in what we have previously observed (ref. 15) and what we have written here - the growth in cell volume is co-occurring with an increase in ribosome content. The only explanation is that SAR11 regulates their number of ribosomes - the process of which is unknown. Nevertheless, we rephrased our statement:

In fact, we have previously shown the ability of SAR11 to modify their ribosome content according to growth rates.

Lines 286-287. While this could explain why there are so many more zombie cells than infected SAR11 it should be clarified throughout the manuscript when comparing total zombies and fraction of infected SAR11. The rest of the manuscript implies it is all about SAR11, by omission of this discussion.

We fully agree and have updated the manuscript accordingly (please see our comments above). Thank you for the valuable feedback!

Lines 407-409. defense mechanisms are easily lost in MAG production because they often have different tetranucleotide frequencies than the underlying core genome and also not all members of a species have them.

Should add a caveat about this.

From the 172 publicly available *Ca. Pelagibacter* ubiquitous reference genomes (18 putative defense mechanisms), six are derived from MAGs (of which one has a putative defense mechanism). The remaining reference genomes are SAGs or genomes of isolates, which are not prone to assembly biases.

Regarding our environmental MAGs, they originate from long-read metagenomes. As long-read metagenomes are expected to be less prone to assembly biases than short-read metagenomes as outlined above, we do not expect a noticeable effect here.